# Qualitative Analysis of Primary Care Provider Prescribing Decisions for Urinary Tract Infections

**DOI:** 10.3390/antibiotics8020084

**Published:** 2019-06-19

**Authors:** Larissa Grigoryan, Susan Nash, Roger Zoorob, George J. Germanos, Matthew S. Horsfield, Fareed M. Khan, Lindsey Martin, Barbara W. Trautner

**Affiliations:** 1Department of Family and Community Medicine, Baylor College of Medicine, Houston, TX 77098, USA; sgnash@bcm.edu (S.N.); Roger.Zoorob@bcm.edu (R.Z.); George.Germanos@bcm.edu (G.J.G.); Matthew.Horsfield@bcm.edu (M.S.H.); fkhan@bcm.edu (F.M.K.); 2Center for Innovations in Quality, Effectiveness and Safety (IQuESt), Michael E. DeBakey Veterans Affairs Medical Center, Houston, TX 77030, USA; Lindsey.Martin@bcm.edu (L.M.); trautner@bcm.edu (B.W.T.); 3Section of Health Services Research, Department of Medicine, Baylor College of Medicine, Houston, TX 77030, USA

**Keywords:** antibiotic stewardship, antibiotics, fluoroquinolones, guidelines, urinary tract infections

## Abstract

Inappropriate choices and durations of therapy for urinary tract infections (UTI) are a common and widespread problem. In this qualitative study, we sought to understand why primary care providers (PCPs) choose certain antibiotics or durations of treatment and the sources of information they rely upon to guide antibiotic-prescribing decisions. We conducted semi-structured interviews with 18 PCPs in two family medicine clinics focused on antibiotic-prescribing decisions for UTIs. Our interview guide focused on awareness and familiarity with guidelines (knowledge), acceptance and outcome expectancy (attitudes), and external barriers. We followed a six-phase approach to thematic analysis, finding that many PCPs believe that fluoroquinolones achieve more a rapid and effective control of UTI symptoms than trimethoprim-sulfamethoxazole or nitrofurantoin. Most providers were unfamiliar with fosfomycin as a possible first-line agent for the treatment of acute cystitis. PCPs may be misled by advanced patient age, diabetes, and recurrent UTIs to make inappropriate choices for the treatment of acute cystitis. For support in clinical decision making, few providers relied on guidelines, preferring instead to have decision support embedded in the electronic medical record. Knowing the PCPs’ knowledge gaps and preferred sources of information will guide the development of a primary care-specific antibiotic stewardship intervention for acute cystitis.

## 1. Introduction

Antimicrobial resistance is increasingly being recognized as one of the major threats to human health globally [1]. The 2015 United States (U.S.) National Action Plan for Combatting Antibiotic Resistant Bacteria set a goal to reduce inappropriate antibiotic use by 50% in outpatient settings [2]. Primary care providers (PCPs) constitute the largest proportion of antibiotic prescribers in the U.S.; therefore, engaging these practitioners is essential to improving outpatient antibiotic stewardship [3]. In 2011, outpatient healthcare providers in the U.S. prescribed 262.5 million courses of antibiotics, with the largest proportion (24%) being prescribed by family medicine physicians [3].

Most studies in outpatient settings have addressed implementing antibiotic stewardship for upper respiratory infections, typically viral infections in which antibiotics are not indicated [4,5]. Few studies have addressed inappropriately prescribing for urinary tract infections (UTI) [6], one of the top three diagnoses for which antibiotics are prescribed in outpatient settings [7]. The first-line agents recommended by the updated 2010 Infectious Diseases Society of America (IDSA) guidelines to treat uncomplicated cystitis are nitrofurantoin, trimethoprim-sulfamethoxazole, and fosfomycin [8]. However, inappropriate choices and durations of antibiotic therapy for UTI are a common problem in primary care [9,10,11]. Providers often choose fluoroquinolones as a first-line agent for uncomplicated cystitis and prescribe antibiotics for longer than the guidelines’ recommended treatment duration [9,10]. Fluoroquinolone use is one of the factors that has led to the emergence and spread of multidrug-resistant *Escherichia coli (E.coli)* strain sequence type 131 [12]. In addition, the U.S. Food and Drug Administration (FDA) issued black box warnings for fluoroquinolones in 2016 [13] and 2018 [14,15] that fluoroquinolones should not be used for uncomplicated UTIs in patients who have other treatment options. These FDA warnings are based on studies indicating an association between fluoroquinolone use and tendonitis and tendon rupture [16,17], QT-prolongation [18], dysglycemia [19,20], neuropathy [21,22,23], and potentially aortic rupture [15]. 

Unfortunately, the 2010 IDSA guidelines had little impact on national outpatient national antibiotic prescribing practices for UTI [24], with fluoroquinolones as the most commonly prescribed antibiotic class in young women both before and after the release of the guidelines [24]. In order to design an effective antibiotic stewardship intervention for outpatient UTI management, we first need to understand why primary care providers are choosing certain antibiotics or certain durations of treatment and what sources of information they rely on to guide their antibiotic-prescribing behavior. To acquire these insights, we used qualitative semi-structured interviews to explore primary care provider prescribing decisions for UTI.

## 2. Results

We interviewed 18 primary care providers working in two private academically affiliated family medicine clinics. We reached data saturation after 13 interviews with our homogeneous sample [25,26].

Our thematic analysis identified seven themes related to providers’ prescribing decisions for acute cystitis (Figure 1). The themes were mapped to the Cabana framework for understanding the barriers to physician adherence to clinical practice guidelines [27]. Four of these themes reflected factors influencing antibiotic prescribing decisions for acute cystitis, two themes described the sources of information they rely upon to guide antibiotic-prescribing decisions, and one theme explored the perceptions of antibiotic resistance in providers’ practices.

### 2.1. Theme 1: Perceived Effectiveness and Efficiency were Common Reasons for Choosing Fluoroquinolones

Most providers reported that they would prescribe trimethoprim-sulfamethoxazole (TMP-SMX) or nitrofurantoin in the case scenario; however, the treatment duration suggested was sometimes longer than recommended by the guideline. Specifically, the preferred treatment duration of nitrofurantoin was 7 days instead of the guideline recommendation of 5 days among approximately half of the providers who selected this medication as their first choice for treatment. The suggested duration of TMP-SMX was 5 days instead of the guideline recommendation of 3 days among a few providers who selected this medication as a first choice, though most providers correctly chose a guideline concordant with the 3-day duration. Relatively few providers chose fluoroquinolones in the case scenario, though a third listed fluoroquinolones second among their top three choices of antibiotics most frequently prescribed for uncomplicated cystitis in response to a different question. In all quotations below, alias initials of providers were included to indicate that the quotations presented come from a variety of participants. 

Common reasons given for why providers might choose fluoroquinolones included prescribing familiarity, “I think a lot of it has to do with familiarity and comfort.” (DS); perceived effectiveness, “No matter what kind of bacteria it is, it would be responsive to Cipro.” (WC); short course of treatment, “In most instances, I just choose Cipro, and it’s easy; it’s for three days.” (WC); broad-spectrum coverage, “Ciprofloxacin is a very strong antibiotic, and it is broad-spectrum.” (JG); and previous patient experience, “If the patient had reported that Cipro had worked well before for them in the past, that might be a consideration.” (SL). Providers also expressed confidence in the ability of ciprofloxacin to kill a wide range of bacteria, reporting, for example, “Physicians don’t want the patient to have a treatment failure and come back in for a second antibiotic.” (AA).

### 2.2. Theme 2: Factors Influencing Providers’ Choice of Antibiotics and Treatment Duration

Providers described multiple considerations when choosing the best antibiotic for acute cystitis. These considerations included allergies, patient sex, pregnancy, patients’ past experience with the antibiotic, familiarity with the antibiotic, previous antibiotic susceptibilities, older age, shorter treatment duration and better compliance, frequency of UTI, presence of diabetes, and cost of antibiotics. A few providers mentioned the potential effect of fluoroquinolones on tendons. For example, one provider noted, “There is concern about the fluoroquinolones and their effect on tendons, but I don’t know whether it’s a media sort of blown up case of concern or whether it’s real.” (MB). Another provider described various factors affecting their choice of antibiotic for particular patients, “Age, whether or not they have frequent UTI, maybe information from their previous urine cultures which shows sensitivities.” (AA).

When the interviewer asked probing questions about the treatment of older patients, some providers mentioned that they would be more likely to choose ciprofloxacin and a longer duration of treatment, while others mentioned that the management of UTI would not change. Similarly, when questioned about recurrent UTIs and the management of diabetic patients with UTI, some providers reported that they would go with “a stronger, broad-spectrum antibiotic such as ciprofloxacin” (BC) and a longer treatment duration. When questioned about male UTIs, providers emphasized the need to rule out a sexually transmitted infection and recommended a treatment duration of up to 14 days. 

### 2.3. Theme 3: Providers’ Mixed Assessments of Nitrofurantoin

Few side effects, low cost, and good tolerance were mentioned by the majority of providers as the benefits of nitrofurantoin, with providers giving such responses as, “Nitrofurantoin, it is one of my “go to”. It is a good medication for uncomplicated cystitis.” (JQ). However, many providers also mentioned that nitrofurantoin is not as “strong” and not as “quick” as fluoroquinolones. One provider explained, “We have a patient population that wants their symptoms to be resolved immediately. Therefore, nitrofurantoin is not a good choice as it is bacteriostatic, and I want to make sure the patient is satisfied and just want to give them something that will resolve symptoms faster.” (BC). 

A few providers were concerned about birth defects based on published case-control studies [28,29], while others thought nitrofurantoin is much safer than TMP-SMX or fluoroquinolones for pregnant women with UTIs, reporting, for example, “It’s one of the ones I use the most, especially in women of childbearing age.” (AV). Provider-reported drawbacks included concerns about lung problems with prolonged use, “When I was on the pulmonary rotation, they were talking about the risk of nitrofurantoin causing pulmonary fibrosis, and I’ve never forgotten that.” (JS), as well as contraindication in patients with low creatinine clearance or in older patients in general, “It’s on the Beers list.” (AA). 

### 2.4. Theme 4: Unfamiliarity with Fosfomycin as a Treatment Option

Most providers were unfamiliar with fosfomycin as a possible treatment option for cystitis, although it is a first-line agent per guidelines [8]. Providers gave such responses as, “I don’t use fosfomycin. I’ve read about it maybe once or twice.” (AV) and “I don’t think one can use it in the United States. I’ve never seen anyone try it, and I’ve never looked into it.” (WC). One provider reported having had a patient who had been prescribed fosfomycin by an infectious disease specialist.

### 2.5. Theme 5: Reliance on Easily Accessed Sources for Clinical Decision-making

In terms of sources of support for clinical decision-making, few providers directly relied on guidelines. Instead, many reported that they use clinical decision support resources, such as UpToDate, accessed through the institution’s electronic health record, “I use UpToDate. That’s simple. I can click from Epic, so it’s right there on my chart. [I’m] talking to the patient, and I just click and pop it up.” (AZ) or through a mobile phone app for staying informed about new information on antibiotics, “I use UpToDate. I have the app on my phone, so I am always searching.” (BC). A few providers mentioned publications such as *Prescriber’s Letter* and resources available from the American Academy of Family Physicians.

### 2.6. Theme 6: Lack of Specific Recall of Guidelines on Treatment of Uncomplicated Cystitis

Only two providers mentioned the IDSA guidelines, though most respondents reported varying degrees of familiarity with unspecified guidelines, saying, for example, “I think I am reasonably up-to-date.” (SC), “I guess I would be fairly familiar.” (JQ), and “It’s been about 10 years since I reviewed the guidelines.” (AV). A few providers acknowledged limited recall of the details of any guidelines regarding the treatment of uncomplicated cystitis, as indicated in the following statements: “Not very. There’s very little reason in something as simple as acute cystitis for me to go searching out the guidelines.” (MB), “Try to stay away from Cipro first line. That’s about all I remember.” (DT), and “I have to admit I’m not particularly conversant. I will usually look at UpToDate and just refresh my memory.” (JS).

### 2.7. Theme 7: Differing Perceptions of Antibiotic Resistance as a Problem in Providers’ Own Practices

When discussing the extent to which antibiotic resistance is a problem in their own practices, providers had widely differing opinions, ranging from little or no resistance, “I have never noticed resistance within my practice.” (GS), to clear, affirmative responses, “Yes, based on cultures, I have had patients who have had organisms identified that have a pretty concerning resistance profile.” (DS). Some providers expressed uncertainty regarding the extent of antibiotic resistance, for example, “It is so hard for me to predict what’s going to be resistant.” (EJ), while others indicated a need for an antibiogram. One provider stated, “I don’t know if there’s actually resistance in this area or not. It would be nice to get, like, a report to say that.” (AA).

## 3. Discussion

In our study PCPs’ antibiotic choices and durations in the case scenario of acute uncomplicated cystitis were generally in line with the guidelines’ recommended first-line therapy. Most providers correctly chose TMP-SMX or nitrofurantoin for the treatment of uncomplicated cystitis in the case scenario. However, ciprofloxacin was often the second antibiotic mentioned when providers responded to the question about the top three most frequently prescribed antibiotics for uncomplicated cystitis. A number of providers chose treatment durations longer than the guideline recommendations. In particular, many were unaware that the current recommended duration of therapy for nitrofurantoin has been reduced to 5 days [8] compared with the 7-day duration in the previous 1999 IDSA guideline [30].

We found that many PCPs believe that fluoroquinolones are more likely to achieve rapid and effective control of UTI symptoms than TMP-SMX or nitrofurantoin. This finding is in line with a previous qualitative study on the effects of knowledge, attitudes, and practices of PCPs on antibiotic selection [31]. Belief that broad-spectrum antibiotics may be more likely to cure an infection was the main reason for a lack of adherence to the guidelines [31]. However, clinical trials have shown that fluoroquinolones are comparable with first-line recommended agents in both clinical and microbiological efficacy in UTIs [32]. Although fluoroquinolones are highly efficacious for uncomplicated cystitis if the uropathogens are susceptible, increasing resistance rates may hamper the effectiveness of fluoroquinolones for empirical antibiotic use. Fluroquinolone resistance increased dramatically in recent years, with >10–30% of community-associated Enterobacteriaceae being quinolone resistant in many parts of the United States and with even higher rates (>50%) in other parts of the world [33]. Furthermore, fluoroquinolones have been moved to the last class of agents (after β-lactams) and should be used only when no other oral options are available because of three warnings from the U.S. FDA that the risk for serious harms outweighs the benefits for patients with uncomplicated UTIs [13,14,15]. Of note, the initial warning preceded our interviews by approximately a year. 

In our study, PCPs’ prescribing decisions took into account multiple factors in determining the best course of treatment for patients with acute cystitis such as pregnancy, patients’ past experience with a specific antibiotic, previous antibiotic susceptibilities, frequency of UTI, older age, and presence of diabetes. Some PCPs chose a longer duration of treatment when the patient was older or had diabetes. These results are consistent with our previous observational database study, performed in the same setting, showing that older age and the presence of diabetes were independent, significant predictors of a longer treatment duration for UTIs [10]. Older women were also more likely to receive fluoroquinolones for uncomplicated UTIs in a study on national antibiotic prescribing trends in outpatient settings in the U.S. [9]. Advanced age, diabetes, and recurrent UTIs may be cognitive biases that drive inappropriate antibiotic use for uncomplicated UTI. Treatment recommendations for otherwise healthy older women, with or without diabetes, are similar to those for younger women [34,35,36]. Similarly, antibiotic choice and duration for sporadic and recurrent UTIs are not different. Broadening the antibiotic spectrum or lengthening the treatment course have not been proven efficacious and elicit concern for potential harm to the individual and the community [37]. 

Another condition that triggered uncertainty for choosing the right antibiotic among the PCPs was UTI in pregnancy. Some were concerned about the association between the use of nitrofurantoin and birth defects reported in case-control studies [28,29]. However, no relationship between nitrofurantoin use and birth defects was observed in a recent large cohort study [38]. An alternative first-line antibiotic that can be used in pregnancy is fosfomycin. However, PCPs in our study were not familiar with fosfomycin as a possible treatment option for cystitis, although it is a first-line agent per IDSA guidelines. 

The fact that our providers did not seem to be aware of fosfomycin as a first-line recommended antibiotic for uncomplicated UTI was not surprising. Fosfomycin was not prescribed in any visits for UTIs in our previous database study [10], and it was prescribed in less than 0.1% of all visits in a recent study of national antibiotic prescribing practices for UTIs in the U.S. [24]. Some of the reasons may be that susceptibility to fosfomycin is not included in the reports received by the providers from the lab, as well as insurance barriers and higher cost of fosfomycin compared with other UTI-relevant antibiotics.

Primary care physicians increasingly have multiple and sometimes conflicting guidelines [39] for patient care and are challenged to maintain the knowledge base represented by the array of guidelines for common problems. In a nationally representative sample of family physicians, Wolff and colleagues [40] found that time constraints, concern for patient well-being, and consideration of particular circumstances were all potential obstacles to guideline adherence; such concerns (e.g., compliance concerns, patient preferences, and previously reported side effects) were noted by the physicians interviewed for the current study. Likewise, nearly half of our study sample thought it would be useful to have guidelines as a component of an electronic medical record, implying that speed and ease of access are important to them. Most providers interviewed were unable to quote from guidelines presented in monograph form but reported routinely accessing various electronic media (e.g., UpToDate or Family Practice Notebook) for prescribing support.

Although most of the primary care providers were concerned about antibiotic resistance in their practice, some did not consider antibiotic resistance to be a problem in their own practice, citing greater resistance problems in inpatient settings or in non-UTI conditions. Likewise, in a Swedish qualitative study, providers were aware of antibiotic resistance but thought of it as a national problem that does not concern their own practice [41]. Some of our interviewed providers indicated a need for an antibiogram specific to their setting to guide their empiric choices and a tool that can help predict the risk of resistance in a specific patient. Unfortunately, an outpatient-specific antibiogram is not available for the participating clinics, and we suspect that few settings have an outpatient-specific antibiogram [42,43].

The main limitation of our study is the relatively small sample size. However, we performed this qualitative study in a preparatory phase of a stewardship intervention and included all PCPs working in two clinics. As with any qualitative study, the results are not meant to be generalizable. In particular, our findings may not be generalizable to public clinics or clinics that are not academically affiliated. Our findings represent an in-depth exploration of the attitudes and beliefs of the study sample. However, findings are transferable as the issues we identified have been seen in other settings [9,31,41]. We are using these findings to develop provider-focused materials for a subsequent clinic-based intervention designed to support providers in their efforts to reduce the risks of drug resistance, to minimize serious side effects, and to enhance overall patient care.

Our findings highlight the necessity of providing point-of-care information for primary care providers from a trusted and easy to access source. This information will help providers to choose the right antibiotic with the right duration, a fundamental aspect of antimicrobial stewardship for UTI.

## 4. Materials and Methods 

### 4.1. Ethics

The study was conducted in accordance with the Declaration of Helsinki, and the protocol was approved by the Ethics Committee of Baylor College of Medicine on 7 February 2019 (H-38265).

### 4.2. Setting and Participants

We selected individual interviews as the data collection strategy most appropriate for our research objectives of exploring individual prescribing decisions. Individual interviews allow for in-depth data collection and minimize the potential influence of group interaction or seniority and supervisory relationships that could have been introduced with focus groups. In addition, individual interviews allowed a much greater flexibility with scheduling, making it more feasible to reach and engage participants. We conducted brief (30 minutes or less), individual, semi-structured interviews with 18 primary care providers (15 family medicine physicians and three physician assistants) in two private, academically affiliated family medicine clinics in a large urban area between late July 2017 and early November 2017. Both clinics utilize Epic Electronic Health Records. The characteristics of our study sample are described in Table 1.

On average, 19,777 patients attend 3248 appointments at these clinics each month. Patients in both clinics are predominantly female (58%) and white (54%). Fluoroquinolones were the most common antibiotic class prescribed (51.6%) for uncomplicated UTI, and 75% of all prescriptions were longer than the recommended duration in these clinics in our previous study in the period of 2011–2014 [10].

### 4.3. Interview Guide Development

The concepts included in our interview guide were based on the Cabana framework for understanding barriers to physician adherence to clinical practice guidelines [27]. According to this framework, three factors affect physician compliance with evidence-based guidelines: knowledge, attitudes, and external factors. Our interview guide (Figure 2) focused on the following issues: awareness and familiarity with the guidelines (knowledge), acceptance and outcome expectancy (attitudes), and external barriers. To assess awareness and familiarity, we included the following questions: “How familiar are you with any guidelines on the treatment of uncomplicated cystitis?”, “What do you recall about these guidelines?”, and “Do these guidelines apply to your practice?”. To assess the acceptance and outcome expectancy, we asked probing questions about the treatment of recurrent UTIs and situations in which the patients were older, diabetic, or male. To help elucidate external factors that might be driving treatment decisions, we included a clinical vignette about a patient who had a diagnosis of acute uncomplicated cystitis and asked the participants to explain their rationale for antibiotic choice and treatment duration and some reasons other primary care providers might prescribe a fluoroquinolone for this condition.

The interview guide also included questions about clinical and nonclinical factors influencing antibiotic choice and treatment duration, such as patient comorbidities or expected patient compliance. We asked open-ended questions about providers’ experience with the specific antibiotics and their preferred sources of support for clinical decision making. We also asked whether they perceived antibiotic resistance to be a problem in their practices. Before starting the study, we pilot-tested our interview guide with two primary care providers at another institution and adapted the guide as per their suggestions.

### 4.4. Interview Procedures

Primary care providers were recruited by the research team during a weekly clinic meeting and provided an overview of the study. After these meetings, individual follow-up recruitment emails were sent by the research interviewer, a doctoral level social psychologist with substantial experience in conducting qualitative interviews (SN). If providers were interested in participating, the interviewer negotiated a convenient time for the session before or after regularly scheduled patient care responsibilities. At the meeting, the interviewer responded to any additional questions or concerns and obtained consent to participate. All invited providers agreed to participate and received a U.S. $20 gift card for their participation. Each individual participated only once.

### 4.5. Data Collection and Analysis

To protect participant confidentiality, the interviewer asked each participant to select an alias, which we designated by initials during the interview and in the resulting transcripts. The interviews were digitally recorded and transcribed verbatim by professional medical transcriptionists. The interviewer listened to all recordings to verify the accuracy of the transcripts before entering them into NVivo 10 software, a qualitative analysis package (QSR International, 2014) that facilitates the management of qualitative data. Based on a preliminary review of the transcripts and the aims of the study, we determined that the interview questions and subsequent probes provided an appropriate outline for the initial identification of key themes [44,45].

Two authors (LG and SN) met regularly to read, code, and discuss the transcribed interviews, maintaining a shared consensus on the coding process. We elected to focus on the explicit content of providers’ reported experiences and perspectives [46]. Overall, we followed the six-phase approach to thematic analysis described by Braun and Clark [47], a sequential but flexible set of guidelines that encompasses familiarization with the data; initial coding; searching, reviewing, and defining themes; and finally, integrating the analytic narrative and data extracts by incorporating illustrative quotes from providers into the description of the final themes [44,47]. Although our sample size was limited to the total number of providers in the clinic setting, we judged that the last 5 to 6 interviews coded were redundant to those previously coded, in repetition without an expansion of the themes, and were confident that data saturation for this sample had been reached [25].

## 5. Conclusions

Our qualitative analysis found that many PCPs believe that fluoroquinolones achieve more rapid and effective control of UTI symptoms than trimethoprim-sulfamethoxazole or nitrofurantoin. The majority of the providers were unfamiliar with fosfomycin as a possible treatment option for cystitis, although it is a first-line agent per guidelines. For recurrent UTI, older patients, or those with diabetes, some providers reported that they would go with “a stronger, broad-spectrum antibiotic such as ciprofloxacin” and longer treatment duration. For support in clinical decision making, few providers relied on guidelines, preferring instead to have decision support embedded in the electronic medical record. Understanding primary care providers’ clinical challenges and knowing their preferred sources of information will enable us to develop a primary care-specific antibiotic stewardship intervention for acute cystitis.

## Figures and Tables

**Figure 1 antibiotics-08-00084-f001:**
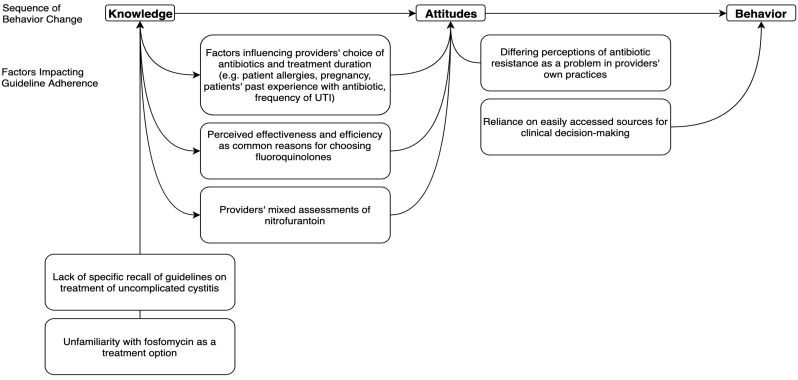
Mapping of the identified themes to the Cabana framework.

**Figure 2 antibiotics-08-00084-f002:**
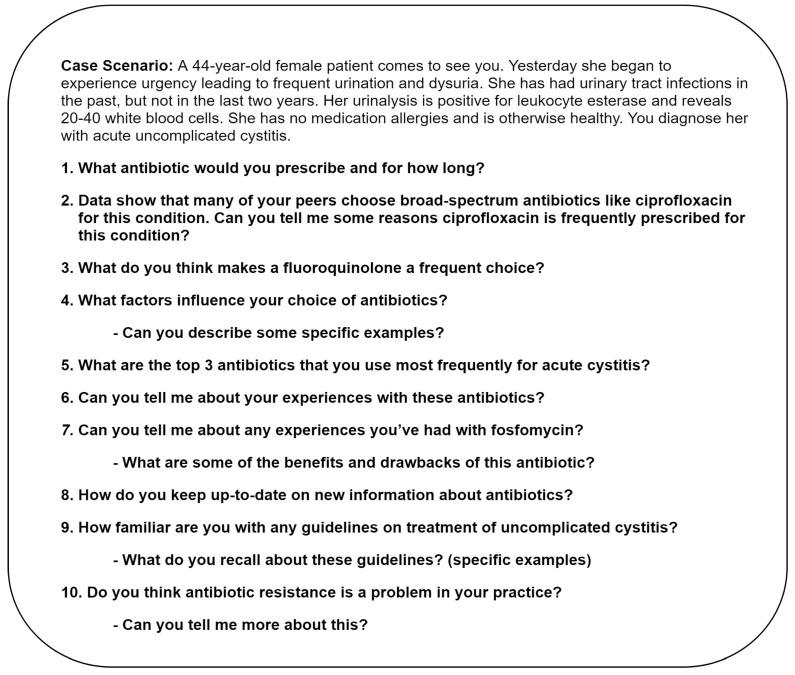
Sample interview questions.

**Table 1 antibiotics-08-00084-t001:** The characteristics of the primary care providers.

Characteristic	*n* (%)
Female	9 (50)
Provider type	
Physician	15 (83)
Physician assistant	3 (17)
Years in practice	
Fewer than ten years	9 (50)
10–20 years	3
Over 21 years	6
Board certified in Family Medicine	14
Board certified in other specialty	1 ^a^
Race/ethnicity	
White	6
Hispanic/Latino	3
Black	1
Asian	8

^a^ One physician was board certified in Internal Medicine.

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
