# Peer review of "Qualitative Analysis of Primary Care Provider Prescribing Decisions for Urinary Tract Infections"

_antibiotics, 2019, doi:10.3390/antibiotics8020084_

Round 1

Reviewer 1 Report

This is a well written and interesting qualitative study to understand the prescription of antibiotics by primary care providers in urinary tract infections. I have only a few comments.

Please change the title, removing the sentence  “We Have a Patient Population that Wants Their Symptoms to be Resolved Immediately”.

In the abstract, the aim of the study should be clear. It must be clear that the study focuses only on the prescription decisions of antibiotics for treatment of urinary tract infections.

The setting of the study were two private, academically-affiliated family medicine clinics. Are these kind of clinics, and physicians working there, representative of primary care service? Why the authors did not make some interviews in public not academically-affiliated clinics? Although it is a qualitative study whose objective is to explore the participants' perception and practice, it seems to me that there may be differences between the private sector and the public sector, as well as among academically-affiliated and not academically-affiliated clinics. It is important to discuss this in the discussion section, maybe as a limitation of the study.

In the methods section was mentioned the use of a guide during the interviews. To better understand the methodology used during interviews it will be important to add this guide as supplementary material. Why individual interviews and not other qualitative methodology (like focus group), considered more adequate for in-depth exploration of attitudes and beliefs? A clear argumentation for the choice of using individual interviews should be added in methods section.

Author Response

Victoria Liu

Assistant Editor

Antibiotics

REF: 516450

Dear Editor and Reviewers,

Thank you for reviewing our manuscript and inviting us to revise our paper. We appreciate your insightful comments and have modified the text according to your suggestions. All revisions were made using track changes in the manuscript. Please find below our replies to your comments.

Comments from Editor

Academic Editor Comments

Before I offer that this manuscript to be sent out for formal reviews, please justify your sample size. It seems small to me and therefore has limited generalisability and applicability.

Our sample was purposively selected and not intended to provide a representative sample of primary care providers. Qualitative studies focus on transferability of the findings versus generalizability, meaning that the qualitative findings can be applied rather than generalized to other settings.1

Qualitative studies, given the depth and richness of information gathered, generally have smaller sample sizes than quantitative studies. Saturation of the data can be achieved with as few as 12 interviews if the sample size is homogeneous.2 Our sample was highly focused and we only interviewed primary care providers working in very similar environments. Both clinics are private, academically affiliated family medicine clinics in large urban area. We reached data saturation after 13 interviews with our homogenous sample. We have clarified this in the first paragraph of our Results section. Please see lines 70-72.

Reviewer 1 comment

This is a well written and interesting qualitative study to understand the prescription of antibiotics by primary care providers in urinary tract infections. I have only a few comments.

Please change the title, removing the sentence “We Have a Patient Population that Wants Their Symptoms to be Resolved Immediately”.

We appreciate the reviewer’s positive comments on our manuscript. We are happy to remove the quote from the title and will leave this decision to the Editor.

In the abstract, the aim of the study should be clear. It must be clear that the study focuses only on the prescription decisions of antibiotics for treatment of urinary tract infections.

As the reviewer suggested, we have clarified the focus of the study in the abstract. Please see lines 22-23.

The setting of the study were two private, academically-affiliated family medicine clinics. Are these kind of clinics, and physicians working there, representative of primary care service? Why the authors did not make some interviews in public not academically-affiliated clinics? Although it is a qualitative study whose objective is to explore the participants' perception and practice, it seems to me that there may be differences between the private sector and the public sector, as well as among academically-affiliated and not academically-affiliated clinics. It is important to discuss this in the discussion section, maybe as a limitation of the study.

We agree with the reviewer and have added this excellent point in the Discussion section. We included only two private academically-affiliated family medicine clinics because our previous quantitative study3 performed in these clinics found low concordance with the guidelines for treatment of UTI. We performed this qualitative study in preparatory phase of a stewardship intervention and included all primary care providers working in these clinics. However, we agree with the reviewer that our findings may not be generalizable to public clinics or clinics that are not academically affiliated. We have added this point in our Discussion section. Please see lines 249-250. We are currently seeking funding to perform similar work in public clinics.

In the methods section was mentioned the use of a guide during the interviews. To better understand the methodology used during interviews it will be important to add this guide as supplementary material. Why individual interviews and not other qualitative methodology (like focus group), considered more adequate for in-depth exploration of attitudes and beliefs? A clear argumentation for the choice of using individual interviews should be added in methods section.

As the reviewer suggested, we have added our sample interview questions in a new figure (Figure 2).

We selected individual interviews as the data collection strategy most appropriate for our research objectives of exploring individual prescribing decisions. In addition to minimizing the potential influence of group processes and seniority and supervisory relationships that could be introduced with focus groups, individual interviews allowed much greater flexibility with scheduling, making it more feasible to reach and engage participants. We have added this point in the Methods section of our revised manuscript. Please see lines 264-269.

Reviewer 2 comments

I think a lot of it has to do with familiarity and comfort (DS);”  perceived effectiveness,  “No matter what kind of bacteria it is, it would be responsive to Cipro (WC);” short  course of  treatment;  “In most instances, I just choose Cipro and it’s easy, it’s for three days (WC)……

what are the (DS), (WC)?? Please check for the whole manuscript.

These initials indicate alias of the interview participants.  As we describe in the Methods section (lines 317-318) to protect participant confidentiality, the interviewer asked each participant to select an alias, which we designated by initials during the interview and in the resulting transcripts. We included these alias initials to show that the quotations presented come from a variety of participants. We have clarified this in the Results section of our revised manuscript. Please see lines 94-95.

Authors should represent the results in figures like distribution or pie charts to make the data more clear.

We agree with the reviewer and presented the results of our qualitative study graphically. Please see our new Figure 2 which includes the themes representing barriers and facilitators for physician adherence to practice guidelines based on Cabana framework.4

1.         Tracy SJ. Qualitative Quality: Eight “Big-Tent” Criteria for Excellent Qualitative Research. Qualitative Inquiry 2010;16:837-51.

2.         Guest G, Bunce A, Johnson L. How Many Interviews Are Enough? Field Methods 2006;18:59-82.

3.         Grigoryan L, Zoorob R, Wang H, Trautner BW. Low Concordance With Guidelines for Treatment of Acute Cystitis in Primary Care. Open Forum Infect Dis 2015;2:ofv159.

4.         Cabana MD, Rand CS, Powe NR, et al. Why don't physicians follow clinical practice guidelines? A framework for improvement. JAMA 1999;282:1458-65.

Reviewer 2 Report

I think a lot of it has to do with familiarity and comfort (DS);”  perceived effectiveness,  “No matter what kind of bacteria it is, it would be responsive to Cipro (WC);” short  course of  treatment;  “In most instances, I just choose Cipro and it’s easy, it’s for three days (WC)……

what are the (DS), (WC)?? Please check for the whole manuscript.

Authors should represent the results in figures like distribution or pie charts to make the data more clear.

Author Response

(The authors gave the same response as above.)
